# A Comparison of a Novel Stretchable Smart Patch for Measuring Runner’s Step Rates with Existing Measuring Technologies

**DOI:** 10.3390/s22134897

**Published:** 2022-06-29

**Authors:** Nina Verdel, Miha Drobnič, Jan Maslik, Klara Björnander Rahimi, Giorgio Tantillo, Alessandro Gumiero, Klas Hjort, Hans-Christer Holmberg, Matej Supej

**Affiliations:** 1Department of Health Sciences, Mid Sweden University, 83125 Östersund, Sweden; nina.verdel@miun.se; 2Faculty of Sport, University of Ljubljana, 1000 Ljubljana, Slovenia; miha.drobnic@fsp.uni-lj.si; 3Department of Materials Science and Engineering, Uppsala University, 75121 Uppsala, Sweden; jan.maslik@angstrom.uu.se (J.M.); klara.rahimi@gmail.com (K.B.R.); klas.hjort@angstrom.uu.se (K.H.); 4STMicroelectronics, 20864 Agrate Brianza, Italy; giorgio.tantillo@st.com (G.T.); alessandro.gumiero@st.com (A.G.); 5Department of Health Sciences, Luleå University of Technology, 97187 Lulea, Sweden; integrativephysiobiomech@gmail.com; 6Department of Physiology and Pharmacology, Karolinska Institutet, 17177 Stockholm, Sweden

**Keywords:** smart patch, biomechanics, cadence, stride rate, SINTEC, wearable sensor, validity

## Abstract

A novel wearable smart patch can monitor various aspects of physical activity, including the dynamics of running, but like any new device developed for such applications, it must first be tested for validity. Here, we compare the step rate while running in place as measured by this smart patch to the corresponding values obtained utilizing ‘‘gold standard’’ MEMS accelerometers in combination with bilateral force plates equipped with HBM load cells, as well as the values provided by a three-dimensional motion capture system and the Garmin Dynamics Running Pod. The 15 healthy, physically active volunteers (age = 23 ± 3 years; body mass = 74 ± 17 kg, height = 176 ± 10 cm) completed three consecutive 20-s bouts of running in place, starting at low, followed by medium, and finally at high intensity, all self-chosen. Our major findings are that the rates of running in place provided by all four systems were valid, with the notable exception of the fast step rate as measured by the Garmin Running Pod. The lowest mean bias and LoA for these measurements at all rates were associated consistently with the smart patch.

## 1. Introduction

Biomechanical evaluation of physical activities (e.g., running) is usually performed in a laboratory using treadmills, force plates, and motion capture systems [1]. However, this approach is usually both inaccessible and too expensive for practitioners, emphasizing the need for less expensive methods suitable for outdoor use. In this context, recent advances in wearable technology, which include a variety of devices worn directly on or loosely attached to the individual, have accelerated the development of less obtrusive and more precise and affordable devices designed to monitor a wide range of parameters in athletes exercising in their normal environments. Advances in technology have enabled individual endurance athletes, sports teams, and physicians to monitor player movement [2], workload [3,4], and biometric markers [5,6] to maximize performance and minimize injury. Monitoring these variables can allow for the detection of biomedical fatigue and early intervention to prevent injury during training and competition, and facilitate the development of improved training plans to optimize athlete performance [7]. However, the technology presently available is often bulky and/or uncomfortable to wear, as well as suffering from poor fixation on the body, which reduces accuracy and sometimes even results in complete loss of data, particularly in connection with intense movements.

These disadvantages can be overcome with smart patches, which weigh less, are more comfortable, and do not move as much relative to the skin, resulting in high-quality signals in connection with the many different movements involved in training and competition. Such patches are not necessarily meant to replace other wearables, but rather to provide novel capabilities that can help achieve immense improvements in, e.g., physical activity, exercise, sport, and health care. 

First developed as part of a European Horizon 2020 Project (www.sintec-project.eu, accessed on 15 June 2022), the novel wearable smart patch technology can be used to monitor a variety of parameters during physical activity, including the dynamics of running, thereby providing completely new possibilities for obtaining information. This technology is based on stretchable electronics that allow high-quality monitoring of performance without obstructing movement. The novel soft, stretchable, and sticky material (3S—PDMS) of which they are composed is compliant and comfortable and, at the same time, does not move relative to the skin, even during excessive movement and sweating. Moreover, the tiny size and width of the sensor, along with its flexibility, allow it to be attached to regions of the body that are inaccessible to many other types of sensors.

Running, one of the most common forms of exercise, is associated with a 27%, 30%, and 23% lower risk of dying from all causes, cardiovascular disease, or cancer, respectively [8]. However, running must be performed properly in order to avoid injuries that can lower quality of life, as well as enhance societal health costs. Indeed, 37–57% of all runners experience some type of injury each year and require an alternative type of exercise.

Although running at 180 steps per minute is commonly recommended [9], it is generally believed that better runners use a higher step rate, which reduces their risk of injury [9,10]. For example, increasing one’s preferred step rate can lower the risk of tibial stress fracture [11]. Moreover, step rate is one of the key parameters utilized to describe and assess running technique [12]. 

Running in place, an alternative form of exercise that can be carried out in a small space (such as an apartment) during, e.g., COVID-19 lock-downs or quarantines, enables higher step rates than when running on a treadmill. In addition, running in place, i.e., without a propulsive component to the ground reaction force, can reduce stress. 

In this context, newly developed smart patches can be employed highly effectively to measure the parameters related to running. Of course, any new device designed for sports or clinical applications must first be tested for validity and reliability. Although wearable sensors are currently a leading worldwide trend in fitness and are used by a variety of groups to monitor variables related to health and physical activity [13,14], these devices are unfortunately often marketed with exaggerated claims that have no solid scientific basis, meaning that the data they provide may be unreliable and of little or no benefit [15,16]. 

The major objective of the present investigation was to assess the concurrent validity of a running parameter—step rate—as assessed on the basis of acceleration measured with newly developed smart patch sensor technology. To this end, the step rate provided by the smart patch was compared to that determined employing bilateral force plates, the gold standard, as well as employing other devices commonly used to assess the dynamics of running, i.e., MEMS accelerometers, a 3D motion capture system, and the ‘‘state of the art’’ Garmin running Dynamic Pod.

## 2. Materials and Methods

### 2.1. Participants

Fifteen healthy and physically active volunteers (men and women, age = 23 ± 3 years; body mass = 74 ± 17 kg, height = 176 ± 10 cm) participated in this study. All participants provided written informed consent before participation. The study was pre-approved by the Ethics Committee for Sport at the University of Ljubljana, Slovenia (033-16/2021-2), which adheres to the principles outlined by the World Medical Assembly Declaration of Helsinki. All participants were physically active for at least 5 h per week and had to be free of neurological and noncommunicable chronic diseases. Written informed consent was obtained from the participants to publish this paper.

### 2.2. Study Design

All participants completed three consecutive 20-s bouts of running in place (on force plates), starting with low-, followed by medium-, and, finally, at high intensity, all self-chosen.

### 2.3. Equipment

Each participant was equipped with a smart patch, three Dytran 7300A5 (MEMS) uniaxial variable-capacitance accelerometers positioned perpendicular to each other (sensitivity: 80 mV/g, range: 50 g, mass: 12 g, Dytran Instruments Inc, Chatsworth, CA, USA), two reflective markers on each foot, and Garmin Running Dynamics Pod (Olathe, KS, USA). The smart patch, Dytran MEMS accelerometers, and Garmin Running Dynamics pods were placed in the lower lumbar region, while two reflective markers were placed in the front part of the participants’ shoes (one on the right shoe and one on the left shoe). The markers were recorded by an infrared (kinematic) motion capture system consisting of 6 Oqus cameras (Oqus 7+, Qualisys, Gothenburg, Sweden). In addition, all measurements were obtained while running in place on bilateral force plates (S2P, Science to Practice Ltd., Ljubljana, Slovenia), see Figure 1.

Dytran accelerometers and S2P bilateral force plates were wired to two synchronized 24-bit Dewe-43 analogue-to-digital converters and DewesoftX data acquisition software (Dewesoft d.o.o., Trbovlje, Slovenia). The Oqus cameras were connected to Qualisys Track Manager (QTM, Qualisys, Gothenburg, Sweden), the Garmin Running Dynamic Pod was connected to a Garmin Forerunner 945, and a smart patch was connected to a custom Windows application (Bio2BITWinApp, STMicroelectronics, Milan, Italy).

#### Smart Patch

The smart patch was developed as part of the European Horizon 2020 Project SINTEC and consists of two parts: an electronic module and a passive skin adhesive patch. The module consists of a custom-made circuit board (STMicroelectronics, Milan, Italy) and a power supply. The electronics board is a Bluetooth Low Energy sensor node featuring an optical interface (light-emitting diode and two spectrometers), a temperature sensor (“NTC”-negative temperature coefficient), and a digital triaxial accelerometer. The device is managed by a Microsoft Windows application that streaming data over the RF interface using a manufacturer’s custom-made protocol. In this study, the device was configured as a high-sampling rate accelerometer. A rechargeable lithium-ion button cell battery, LIR2032, with a nominal voltage of 3.6 V and a capacity of 45 mAh was used as a power supply. The battery was adapted to be mounted on a surface. Half of the bottom side of the battery is covered with insulating tape, while on the top side a 5 mm wide copper tape strip is placed, see Figure 2. To provide a connection between the sensor board and the battery and to allow firmware update of the sensor board, a flexible circuit board with an accessible connector (i.e., Molex 0.5 mm Pitch SlimStack Plug and Receptacle connector) has been designed. In addition, the entire assembly is encapsulated as all electrical components are functional, see Figure 3. Two materials are used for encapsulation: a low-friction polyurethane tape (inspire^®^ 2231, Transcontinental AC UK Ltd., Wrexham, UK), and a soft silicone gel (Silbione 4645, Elkem Silicones Scandinavia AS, Oslo, Norway). 

The passive patch (Figure 4) consists of three materials: a stretchable fabric sports tape (Ultra Performance Kinesiology tape, Mueller Sports Medicine Inc., Prairie du Sac, WI, USA) cut into 4.5 cm wide, and 11.2 cm long patches that are large enough to cover the paired modules and an additional space at the edge to ensure secure adhesion to the skin. In the center of the sports tape patch is a thin polyurethane film (inspire^®^ 2150, Transcontinental, Montreal, QC, Canada) that forms a non-adhesive interior. To facilitate the application of the wearable device, two adhesive dots made of adhesive silicone gel (Silbione 45, ELKEM, Oslo, Norway) were placed inside to cover the adhesive area and the gripping points. The patch is attached to a protective liner.

### 2.4. Data Analysis

Data from the Qualisys Oqus cameras and Dytran accelerometers were acquired at a sampling frequency of 1 kHz, whereas data from the force plates and smart patch were acquired at a sampling frequency of 256 Hz, and data from the Garmin Running Dynamic Pod were acquired at 1 Hz. The data from the force plates and the Dytran accelerometers were synchronized because they were both connected to a coupled set of analog-to-digital converters, which were started by a TTL signal coming from the Qualisys motion capture system. The data from the smart patch and the Dytran accelerometers were synchronized by matching the first location of the peak/local maxima. The data from the smart patch and the Garmin Running Dynamic Pod were synchronized by the starting procedure, where participants had to stand still before the measurements and thereafter with time stamps, and manually by overlapping the jumps in the different data sets. 

The Garmin Forerunner 945 provided the step rate from the Garmin Running Dynamic Pod, while with the smart patch and Dytran accelerometers, the step rate was calculated on the basis of the absolute acceleration and raw data provided by the Qualisys system (5). Figure 5a displays a signal (in terms of units of body weight) acquired while running in place on a force plate. The ground reaction force was greatest after the participant’s toes touched the force plate, while it was 0 when the foot was in the air. Figure 5b,d shows the signals from two different accelerometers (the smart patch and Dytran) attached to the pelvis. In this case, acceleration was most rapid after the toe touched the ground and slowest when the toe left the ground and before the second foot (toe) touched the ground. Therefore, the step rate was calculated from the locations of the peak/local maxima [17,18] of acceleration and force as follows:(1)$$\mathrm{Step}\;\mathrm{rate}\;\left(i-1\right)\;\left[\min^{-1}\right]=\;\frac{60}{t_{peak\left(i\right)}-t_{peak\left(i-1\right)}}$$
where *t_peak_* is the time at which the peak occurred and *i* are integers from 2 to the total number of the steps. Figure 5c illustrates the signal acquired with the 3D motion capture system Qualisys. The markers were placed on the front of the participant’s shoes (see Figure 1), and the signal thus represents the vertical distance of the marker above the force plates. If this distance was below a given threshold, the foot was considered to be on the force plate; otherwise, it was in the air. Using the valleys/local minima, the step rate of the markers on the feet (the Qualisys motion capture system) was calculated in a manner analogous to that described above. All calculations were performed in MATLAB software (version R2020b, The MathWorks Inc., Natick, MA, USA).

Data were tested for normality using the Kolmogorov–Smirnov test, as well as the differences between data. Because normality was rejected for all data (*p* < 0.05), statistical tests that do not assume normality were used. Concurrent validity, which evaluates the association between data from a new device and another device considered to be more valid (gold standard), is reported. The Wilcoxon signed-rank test was used to assess systematic bias between trials, with statistical significance set at *p* < 0.05. The mean bias (a systematic difference between the measurements obtained and the true value of the parameter) and limits of agreement (LoA, the interval within which a pre-defined proportion of the differences between measurements lie) were calculated using a nonparametric approach, as proposed by Bland and Altman [19].

## 3. Results

Different measurements of slow, medium, and rapid rates, as well as all three rates are compared in Table 1. As can be seen, the only measurements that differed significantly were the medium and high rates, as well as all rates combined, determined by the Garmin Running Pod.

In addition, the smallest range of the LoA between measurements at medium and fast rates, as well as all rates combined, was demonstrated by the smart patch, while at the slow rate, this range was smallest for measurements with the Dytran accelerometers. The step rate determined by the smart patch is not statistically different from the gold standard force plates or the state-of-the-art Dytran accelerometer and 3D visual system, making Smart Patch a viable alternative for measuring the step rate. The corresponding Bland–Altman plots are presented on Figure 6.

## 4. Discussion

The current investigation indicates that the validity of the step rate while running in place by the novel smart patch is acceptable at all rates examined. However, the reliability of this device could not be determined because the participants chose their three different rates freely; therefore, we could not ensure that repetitions involved the same rates. 

The rate of running in place by our participants in the laboratory was measured not only with the smart patch but also with the state-of-the-art devices the Garmin Running Pod, Dytran accelerometers, and the Qualisys motion capture system, with a comparison of the values obtained with all four approaches to those provided by S2P bilateral force plates, the gold standard. With slow running in place, the mean bias was less than 1 step/min (between −0.4 and 0.2 steps/min) for all four devices—lowest (0.1 steps/min) for the Qualisys motion capture system and highest (−0.4 steps/min) for the Garmin Running Pod. In their investigation of the validity of measuring running cadence (step rate) with earbuds, Nijs et al. [20] observed a maximal mean cadence of 181 steps/min, which is comparable to our slow rate of running in place (179 steps/min). In their study, the mean bias between the values provided by the force plates and the earbuds was 1.3 steps/min, which is higher than the mean bias for any of the devices evaluated here. In another study, the mean bias between the running cadence (step rate) provided by a motion-capture system and a smartwatch was 1.2 steps/min [21], which is also higher than for all four of our devices.

During slow running in place, the LoA was lowest with the Garmin Running Pod (−8.6 to 6.1 steps/min) and highest with the Qualisys motion capture system (−17.6 steps/min to 19.2 steps/min). This rate is comparable to the running cadence (step rate) of recreational runners [22] and there were no statistically significant differences between the values obtained with any of our four devices in comparison with force plates.

Here, the medium and high rates of running in place (224.8 ± 38.3 and 281.2 ± 60.1 steps/min, respectively) were higher than the normal running cadence/step rate (170–190 steps/min [22]). While running in place at a medium rate, the lowest mean bias (0.2 steps/min) was exhibited by the Dytran accelerometers and the highest (−2.5 steps/min) by the Garmin Running Pod, with the values obtained from the latter differing significantly from those obtained with the force plates. Similarly, the LoA was lowest for the Dytran accelerometers (−14.1 to 14.5 steps/min) and highest for the Garmin Running Pod (−24.1 to 13.8 steps/min). With the exception of the mean medium and fast rates obtained with the Garmin Running Pod, none of the values determined with the other devices differed significantly from those obtained with the force plates.

At the fastest rate of running in place, the mean values obtained with the smart patch and force plates were the same, i.e., the smart patch exhibited no bias, whereas bias was highest for the Garmin Running Pod (−38.9 steps/min). Similarly, the LoA was lowest for the smart patch (−19.3 to 18.8 steps/min) and highest for the Garmin Running Pod (−186.6 to 3.8 steps/min), with the values provided by the latter device differing significantly from the force plate measurements.

Overall (i.e., when combining slow, medium, and fast running in place), the mean bias was least for the smart patch (0.2 steps/min) and highest for the Garmin Running Pod (−13.9 steps/min), with the values provided by the latter device again differing from those obtained with the force plates. The Garmin Running Pod was the only device tested that provided measurements of step rate, both at medium and high intensity and overall, that differed significantly from those obtained with the gold standard force plates.

One limitation of the present study is that the rate of running in place was self-selected by the participants, which allowed them to achieve their most rapid step rate but made it impossible for us to determine the reliability of the different devices since the rate may have varied between repetitions. In addition, the standard deviations of the rates measured by all devices were similar and high, indicating a certain degree of inaccuracy in the measurements obtained, as also seen in the Bland–Altman plots. An additional limitation was that the determination of step rate while running in place by the Dytran accelerometers, smart patch, and Qualisys system was based simply on the detection of local minima and maxima [17,18]. Nevertheless, the rates provided by these different devices did not differ statistically, indicating that this approach was sufficiently accurate. 

In conclusion, we demonstrated here that measurements of rates of running in place were similar to rates of running by all four of the devices examined and are valid, whereas at a fast step rate, the values provided by the Garmin Running Pod are no longer valid. The lowest mean bias and LoA in these measurements at all rates were associated consistently with the smart patch.

## Figures and Tables

**Figure 1 sensors-22-04897-f001:**
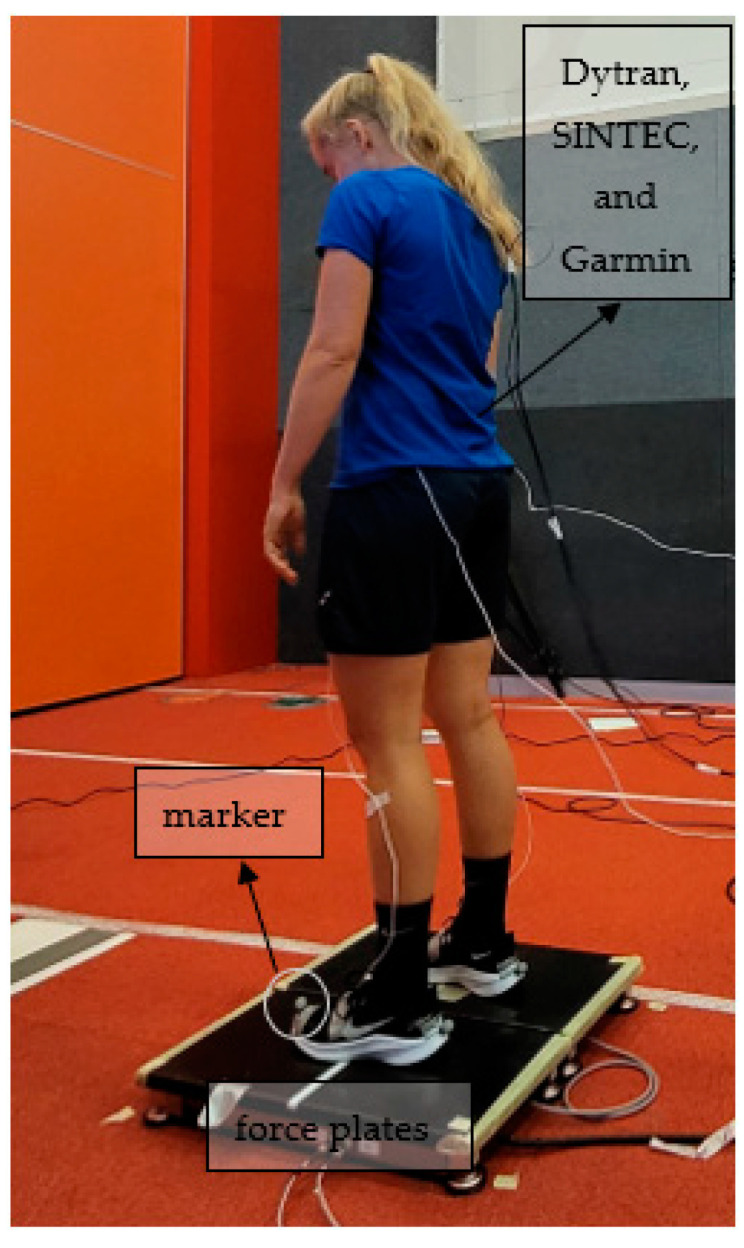
Participants standing on force plates.

**Figure 2 sensors-22-04897-f002:**
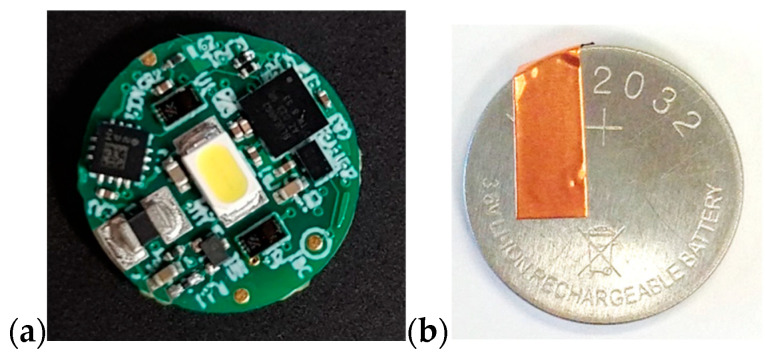
(**a**) Sensor board, (**b**) adapted LIR 2032 battery.

**Figure 3 sensors-22-04897-f003:**
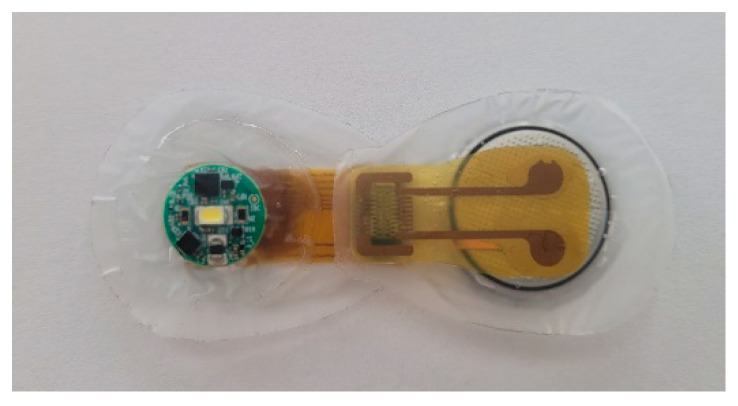
Encapsulated and connected sensor board and battery via Molex connector.

**Figure 4 sensors-22-04897-f004:**
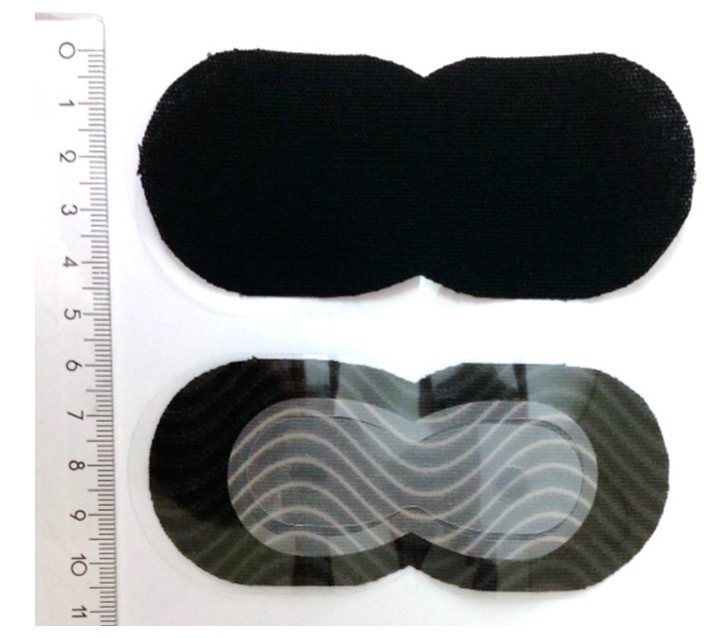
Passive patch.

**Figure 5 sensors-22-04897-f005:**
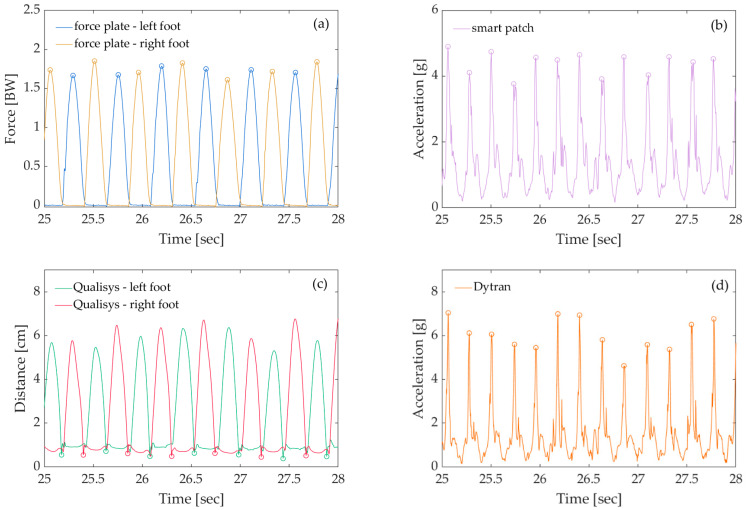
Examples of signals acquired from (**a**) the Qualisys motion capture system, (**b**) the S2P bilateral force plates, (**c**) the Dytran accelerometers, and (**d**) the smart patch. The circles on the graphs represent the peaks and valleys from which the step rate was calculated.

**Figure 6 sensors-22-04897-f006:**
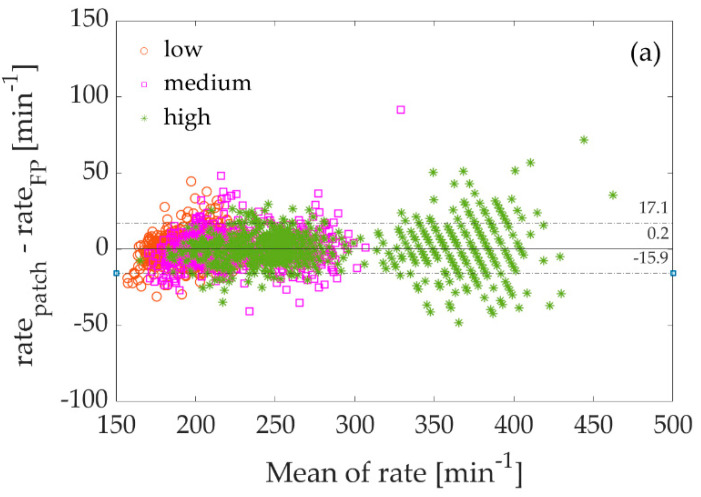
Bland–Altman plots of the rate of running in place as measured with (**a**) the smart patch versus force plates, (**b**) Dytran accelerometers versus force plates, (**c**) the Qualisys motion capture system versus force plates, and (**d**) the Garmin Running Pod versus S2P bilateral force plates (FP). Orange circles = slow running in place; pink squares = medium; green stars = fast. The dashed lines represent the limits of agreement (LoA), while the solid lines depict bias.

**Table 1 sensors-22-04897-t001:** Comparison of the rates of running in place (min^−1^, means ±SD) as determined with S2P bilateral force plates, the smart patch, Dytran accelerometers, the Qualisys motion capture system, and the Garmin Running Pod.

Rate	Force Plates	Smart Patch	Dytran	Qualisys	Garmin	Bias	*p*-Value	Limits of Agreement [min^−1^]
low	197.0 ± 15.6	197.2 ± 17.0	/	/	/	0.2 ± 8.5	0.78	−12.5 to +13.4
	197.0 ± 15.6	/	197.3 ± 17.0	/	/	0.2 ± 8.4	0.98	−12.7 to +13.6
	197.0 ± 15.6	/	/	197.8 ± 26.4	/	0.1 ± 11.7	0.96	−17.6 to +19.2
	195.2 ± 14.6	/	/	/	194.8 ± 16.0	−0.4 ± 5.5	0.84	−8.6 to +6.1
medium	224.8 ± 28.3	225.2 ± 29.3	/	/	/	0.3 ± 10.1	0.87	−14.7 to +15.8
	224.8 ± 28.3	/	225.0 ± 28.8	/	/	0.2 ± 8.7	0.72	−14.1 to +14.5
	224.8 ± 28.3	/	/	225.3 ± 31.1	/	0.5 ± 15.0	0.56	−18.6 to +18.4
	218.6 ± 24.6	/	/	/	216.1 ± 20.3	−2.5 ± 10.2 *	< 0.05	−24.1 to +13.8
high	281.2 ± 60.1	281.2 ± 60.2	/	/	/	0 ± 11.9	0.85	−19.3 to +18.8
	281.2 ± 60.1	/	281.5 ± 61.5	/	/	0.3 ± 15.1	0.92	−23.5 to +24.2
	281.2 ± 60.1	/	/	281.3 ± 60.5	/	0.1 ± 15.9	0.86	−25.4 to +25.5
	263.7 ± 51.0	/	/	/	224.8 ± 28.2	−38.9 ± 64.1 *	< 0.05	−186.6 to +3.8
all	239.1 ± 54.9	239.2 ± 55.2	/	/	/	0.2 ± 10.5	0.88	−15.9 to +17.1
	239.1 ± 54.9	/	239.3 ± 55.7	/	/	0.3 ± 11.5	0.89	−17.6 to +18.3
	239.1 ± 54.9	/	/	239.5 ± 56.5	/	0.4 ± 17.5	0.69	−20.3 to +21.8
	225.8 ± 44.1	/	/	/	211.9 ± 25.4	−13.9 ± 41.5 *	< 0.05	−172.9 to +6.4

* significantly different mean step rate.

## Data Availability

The data presented in this study are available on request from the corresponding author providing the access does not interfere with the conditions provided by the ethics committee.

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
