# Peer review of "A Comparison of a Novel Stretchable Smart Patch for Measuring Runner’s Step Rates with Existing Measuring Technologies"

_sensors, 2022, doi:10.3390/s22134897_

Round 1
Reviewer 1 Report
The paper has a narrow focus on one brand of product in wearable computing, and lacks breadth and depth. There is a lack of context, firstly. There's no contextual background. There really should be an introduction to the field of wearable computing that puts the paper in the broader intellectual landscape. There should also be a general scientific principle, not just a narrow one-brand summary. In the past things changed slowly, so if you for example wrote a research paper on something like the Ampex 350 tape recorder made in the 1960s, it might be valid for some time, and might pass the threshold of interest for academia. But today technology changes so fast that the paper will be obsolete by the time it is published. It does not merit archival in the scientific literature. And even if it did, there's a matter of narrow focus. Look back in the history of science, and if you found something about tape recorders of the 1960s it would be a general treatise in fundamental scientific principles of magnetic hysterisis for example, and not brand-specific.
Secondly the paper lacks scientific merit. There is no fundamental principle behind the work, it is merely a glimpse at one brand of product.
This work should be written as a product review in a weblog somewhere and it should be published on the internet on a website immediately for a short time period until the website goes to 404 not found around the same time the article loses its relevance when next year's model comes in.
The work done here is of narrow ephemeral interest best suited to being a product review on social media for example, and not of scientific interest or merit.
Reviewer 2 Report
Comments
This is an interesting paper that shows a new innovation to measure runner’s running styles and shows how it compares with existing technologies.
The work looks soundly based but more explanation is needed to describe more clearly what the study aimed to do.
Specific comments:
The title needs to be shorter e.g. ‘A comparison of a novel stretchable smart patch for measuring runner’s step rates with existing measuring technologies.’
Page 1, line 37: Can you say more about what running properly means e.g. number of steps per minute? Also what does ‘running in place’ mean? Is it with or without a treadmill?
Page 2, line 43: Possible reword: ‘In addition, if this type of exercise is accompanied by abdominal drawing-in manoeuvres, it increases the thickness…’
Line 58: ‘which are inaccessible…’
Line 64: Would suggest removing words ‘aggressing and. so just say ‘exaggerated claims’.
Line 66: Suggest deleting words ‘the first published’ and just say ‘we present a comparison’.
Line 69: ‘Garmin’
2 Materials and Methods
The paper needs a section that explains the aim of the study and the method more clearly. It should say what variables are being measured, e.g., steps per minute, why this is useful (referencing some literature on running health), and what you are looking for in the results to validate the new technology (smart patch). For example, are you looking for some absolute characteristics in the data, as shown in the charts or table, or are you looking for similarity of results between your technology and existing technology?
In relation to this you could explain what the measures ‘mean bias’ and ‘LoA’ are?
P5. Figure 4 could have some explanation for the layman to guide them on what they are looking at and what the differences in the wave patterns means in terms of data quality.
P6 Table 1 seems to show that the results from the Garmin system at medium and high step rates are significant. Does this mean that for this technology they are different to the other systems.
P6. Would suggest enlarging the 4 charts and putting them into a column so that the shapes )circles, squares, stars) are clearer.
P8, line 259 ‘..were similar to..’
P9-10: The references seem relevant and are compiled and presented in an appropriate format.
Round 2
Reviewer 1 Report
The manuscript is improved with regards to being general (scholarly) rather than brand-specific, i.e. for archival and timelessness purposes.
The review of the wearables field is present but weak.
There is still missing a fundamental scientific basis, e.g. mathematical foundation, or fundamental scientific principles upon which the work is founded.
Reviewer 2 Report
I feel that this paper is much improved and is almost ready for publication.
The only part where further clarification would be useful is at the start of the results section. Since the smart patch is the focus of the paper, it would be useful to make it clearer how this performed e.g. that the results obtained were not significantly different than for the other three systems i.e. so it becomes a viable alternative to them.
